# Response Surface Methodology-Based Optimization of the Chitinolytic Activity of *Burkholderia contaminans* Strain 614 Exerting Biological Control against Phytopathogenic Fungi

**DOI:** 10.3390/microorganisms12081580

**Published:** 2024-08-02

**Authors:** Imen Ben Slimene Debez, Hayet Houmani, Henda Mahmoudi, Khaoula Mkadmini, Pedro Garcia-Caparros, Ahmed Debez, Olfa Tabbene, Naceur Djébali, Maria-Camino Urdaci

**Affiliations:** 1Laboratory of Bioactive Substances, Center of Biotechnology of Borj-Cedria (CBBC), BP 901, Hammam-Lif 2050, Tunisia; imen.debez@cbbc.rnrt.tn (I.B.S.D.); olfa.tabbene@cbbc.rnrt.tn (O.T.); naceur.djebali@cbbc.rnrt.tn (N.D.); 2Laboratory of Extremophile Plants, Center of Biotechnology of Borj-Cedria (CBBC), BP 901, Hammam-Lif 2050, Tunisia; hayet.houmani@cbbc.rnrt.tn (H.H.); ahmed.debez@cbbc.rnrt.tn (A.D.); 3International Center for Biosaline Agriculture (ICBA), Academic City, Near Zayed University, Dubai P.O. Box 14660, United Arab Emirates; 4Useful Materials Valorization Laboratory, National Centre of Research in Materials Science, Technologic Park of Borj Cedria, BP 073, Soliman 8027, Tunisia; khaoulafayrouzahammi@gmail.com; 5Agronomy Department of Superior School Engineering, University of Almería, 04120 Almeria, Spain; pedrogar123@hotmail.com; 6Laboratoire de Microbiologie, Université de Bordeaux-Bordeaux Sciences Agro, UMR 5248, 1 Cours du Général de Gaulle, 33175 Gradignan, France; maria.urdaci@agro-bordeaux.fr

**Keywords:** *Burkholderia cepacia*, chitinase, RSM technique, *Phoma medicaginis*

## Abstract

As part of the development of alternative and environmentally friendly control against phytopathogenic fungi, *Burkholderia cepacia* could be a useful species notably via the generation of hydrolytic enzymes like chitinases, which can act as a biological control agent. Here, a *Burkholderia contaminans* S614 strain exhibiting chitinase activity was isolated from a soil in southern Tunisia. Then, response surface methodology (RSM) with a central composite design (CCD) was used to assess the impact of five factors (colloidal chitin, magnesium sulfate, dipotassium phosphate, yeast extract, and ammonium sulfate) on chitinase activity. *B. contaminans* strain 614 growing in the optimized medium showed up to a 3-fold higher chitinase activity. This enzyme was identified as beta-N-acetylhexosaminidase (90.1 kDa) based on its peptide sequences, which showed high similarity to those of *Burkholderia lata* strain 383. Furthermore, this chitinase significantly inhibited the growth of two phytopathogenic fungi: *Botrytis cinerea* M5 and *Phoma medicaginis* Ph8. Interestingly, a crude enzyme from strain S614 was effective in reducing *P. medicaginis* damage on detached leaves of *Medicago truncatula*. Overall, our data provide strong arguments for the agricultural and biotechnological potential of strain S614 in the context of developing biocontrol approaches.

## 1. Introduction

*Burkholderia* spp. are aerobic Gram-negative bacteria present in soil, water, plant rhizosphere, humans, various animal species, and hospital environments. They are used for biocontrol, bioremediation, and plant growth promotion [1]. Several *Burkholderia* species have shown efficacy against different plant pathogens, including *Colletotrichum gloeosporioides*, *Rhodotorula pilimanae*, *Penicillium digitatum*, *P. expansum*, *Aspergillus flavus*, *Botrytis cinerea*, *A. ochraceus*, *A. alternata*, *Macrophomina phaseolina*, *Ganoderma boninense*, *Fusarium* spp., and *Rhizoctonia solani* [2]. Chitin is present in various organisms such as fungi, sponges, coralline algae, crustacean shells, mollusks, and insects [3,4,5,6]. Chitin oligosaccharides and chitooligosaccharides are known as aminooligosaccharides [7], which, due to their safe solubility and diverse physiological functions, have potential utilization in functional foods, drugs, cosmetics, and agriculture [7]. Enzymatic [8], chemical [9], and physical [10] methods are used for producing aminooligosaccharides from chitinous biomass, with enzymatic methods being preferred for their efficiency, mild reaction conditions, high yield, and minimal pollution compared to chemical and physical methods [11].

Chitinases are categorized as exo-type (EC.3.2.1.52, also known as β-N-acetylhexosaminidase) and endo-type (EC.3.2.1.14) based on cleavage type [7,12]. In the CAZy database, endo-type chitinases are primarily found in glycoside hydrolase (GH) 18 and GH 19, while exo-type chitinases are mainly in GH 18 and GH 20. Most GH 20 family enzymes are β-N-acetylhexosaminidase and target β-N-acetylglucosamine (β-N-GlcNAc) and β-N-acetylgalactosamine (β-GalNAc) units in various substrates like chitosan, chitin, glycosphingolipids, and other glycoconjugates [13]. Chitinases degrade substrates from diverse sources such as bacteria, fungi, insects, plants, animals, and humans. Considering their multiple applications and their economic potential, optimizing chitinase activity using statistical approaches could reduce time and costs in trials, although the classic one-factor-at-a-time strategy is also applied if necessary. Several statistical approaches can be used for experiment optimization [14]. Response surface methodology (RSM) is a valuable statistical approach for studying and improving intricate processes by utilizing quantitative data from a well-thought-out experimental design to establish and solve a multivariate equation [15]. Central composite design (CCD) is a frequently employed response surface design that specifically targets the experimental boundaries of each factor without exceeding them [16]. It allows for the identification of factor combinations that result in an optimal response. Furthermore, significant interactions between variables can be identified and quantified by using this approach [17]. Recent studies have highlighted the pertinence of RSM for analyzing substrate type in order to optimize chitinase production by *Thermomyces lanuginosus* MTCC 9331 [18], *Achromobacter xylosoxidans* [19], *Aspergillus niger* EM77 [20], and *Paenibacillus elgii* PB1 [21]. CCD, a well-established and widely used statistical technique for determining the influence of key factors by a small number of experiments, has been widely used for further optimizations of enzyme production by beneficial microorganisms, such that Lysinibacillus fusiformis B- CM18 [17], *Pseudomonas aeruginosa* FPK22 [22], and *B. cereus* GS02 [23] producers of chitinases that can be used in industry and in the control of plant diseases.

In the present study, *Burkholderia contaminans* S614 was isolated from a soil in southern Tunisia. RSM using a central composite plane CCD was then applied to assess the effects of five factors including colloidal chitin, magnesium sulfate (MgSO_4_·7H_2_O), dipotassium phosphate (K_2_HPO_4_), yeast extract, and ammonium sulphate ((NH_4_)2SO_4_) and their interactions on chitinase activity. Chitinase was identified on the basis of its peptide sequences. Furthermore, from the perspective of its utilization for biocontrol purposes, we assessed the ability of the identified chitinase to (i) inhibit the growth of two important phytopathogenic fungi, *Botrytis cinerea* M5 and *Phoma medicaginis* Ph8, and (ii) to reduce the damage produced by *P. medicaginis* on the leaves of *Medicago truncatula*.

## 2. Materials and Methods

### 2.1. Chitinase-Producing Bacteria

Soil samples were collected from the rhizosphere of an alfalfa oasis located in the Kebili region, southern Tunisia (longitude: 8°57′33.616″ E; latitude: 33°41′48.541″ N). The soil samples were stored at 4 °C for 4 days for microorganism analysis. The soil samples were also examined for their physico-chemical properties in the CRDA (Commissariat Régional de Développement Agricole) of Nabeul. The soil exhibited the following major characteristics: pH 7.8, 6% silt, 10% clay, 84% sand, 0.38 organic matters, 1.7 ppm Fe, 0.38 ppm Zn, and 0.14 ppm Cu. One gram of soil sample was mixed with 5 mL sterile physiological water and filtered after 5 h. Serial dilutions of the soil filtrate were placed on Luria Bertani agar plates (LB) (Sigma-Aldrich Inc., Saint-Louis, MO, USA). After incubation at 25 °C for 2 days, the plates with the greatest number of isolated colonies (between 30 and 100 colonies) were selected. From each plate, one colony representing each morphological type was picked and streaked for purity on LB agar plates.

Bacterial isolates were then plated on colloidal chitin agar plates containing (in grams per liter): (NH_4_)_2_SO_4_, 7; K_2_HPO_4_, 1; NaCl, 1; MgSO_4_·7H_2_O, 0.1; yeast extract, 0.5; colloidal chitin, 5; and agar 15. The pH of the medium was adjusted to 7.2 and it was grown for one week at 30 °C for. Strain S614 showed a defined hydrolysis zone and was selected as an efficient chitinase-producing strain. This strain was grown aerobically on Luria Bertani (LB) broth for 48 h at 30 °C and maintained at −80 °C in 25% (*v*/*v*) glycerol. If necessary, the preserved strain was subcultured on Luria Bertani (LB) agar medium (Sigma-Aldrich Inc., Saint-Louis, MO, USA) and incubated for 48 h at 30 °C then stored at 4 °C before use.

### 2.2. Cell Culture Conditions of S614 Strain

Bacterial strain S614 was grown in minimal medium containing (in grams per liter): (NH_4_)_2_SO_4_, 7; K_2_HPO_4_, 1; NaCl, 1; MgSO_4_·7H_2_O, 0.1; yeast extract, 0.5; and colloidal chitin, 5. The pH of the medium was adjusted to 7.2. The culture was incubated at 30 °C in a rotating incubator at 150 rpm for 8 days. The culture supernatant was collected after centrifugation at 13,000× *g* for 15 min and stored at −20 °C for later analysis.

### 2.3. Isolate Identification

Identification of the bacterium was carried out by 16S rRNA sequencing. PCR amplification of the 16S rRNA gene was performed by using the following primers: forward F27 (5′-AGAGTTTGATCATGGCTCAG-3′) and reverse 1522R (5′-AAGGAGGTGATCCAGCCGCA-3′). The PCR amplification program included an initial denaturation at 96 °C for 120 s, 40 cycles of denaturation at 96 °C for 45 s, annealing at 56 °C for 30 s, and extension at 72 °C for 120 s, and a final extension at 72 °C for 300 s. The resulting 16S rRNA sequence was compared using the National Center for Biotechnology Information (NCBI) nucleotide database. The sequence was deposited in GenBank under accession number (MW922877).

A phylogenetic tree was established using the neighbor-joining (NJ) method of MEGA 11 software to analyze the evolutionary connections based on the 16S rRNA gene sequence of the strain S614 and twelve 16S rRNA references sequences. Matrix distances were calculated based on distance p. The confidence level of each branch was tested by seeding 1000 replicates generated with a random seed.

### 2.4. Colloidal Chitin Preparation and Chitinase Activity Assay

A colloidal chitin solution was made from commercial chitin powder (Sigma-Aldrich Inc., Saint-Louis, MO, USA) according to [24]. Five grams of commercial chitin powder was slowly added into 200 mL of concentrated hydrochloric acid at 4 °C under vigorous stirring. After homogeneous dispersion of chitin powder, the mixture was heated gently up to 37 °C with moderate stirring. The mixture viscosity increased rapidly and then, within a few minutes, began to decrease. To discard the non-dissolved chitin, the mixture was filtered through glass wool. The filtrate was still poured into deionized water at 4 °C and stirred for 30 min, before storing the suspension overnight at 4 °C. The suspension was then decanted and filtered through Whatman no. 3 filter paper. The residue was washed with water until the suspension became neutral. The acid-free residue was resuspended in deionized water with vigorous stirring to prepare the so-called colloidal chitin solution. Storage was performed for a few weeks in the dark at 4 °C.

Chitinase activity was assayed in a 600 μL reaction mixture containing 0.5% (*w*/*v*) colloidal chitin and enzyme solution in 0.1 M sodium acetate buffer, pH 6.0. After incubation at 50 °C for 1 h, the reaction was stopped by boiling at 100 °C for 10 min and the mixture was centrifuged at 12,000× *g* for 5 min to eliminate the remaining chitin. The reducing sugar liberated into the reaction mixture was determined by the 3,5-dinitrosalicylic acid (DNS) method [25] through the recording of the absorbance at 540 nm using a standard curve of N-acetylglucosamine (GlcNAC). One unit (U) was defined as the amount of chitinase able to release 1 μmol of free GlcNAC/min.

### 2.5. Experimental Design

Based on previous data, we noticed that there were eight factors that could influence the activity of chitinase. Using the Plackett–Burman design [24], only 5 factors showed a considerable effect on chitinase. A CCD methodology was used to optimize the effects of the five tested factors, including concentrations of ammonium sulphate (X1), potassium phosphate (X2), magnesium sulphate (X3), colloidal chitin (X4), and yeast extract (g/L) (X5), on the chitinolytic activity of *Burkholderia contaminans* strain 614 (YCA). Each element in the experimental design was tested at five levels (−2.38, −1, 0, 1, and 2.38). This design was used to assess the effects of (NH_4_)_2_SO_4_ concentrations ranging from 7 to 18.83 g/L, K_2_HPO_4_ concentrations ranging from 1 to 2.69 g/L, MgSO_4_ concentrations ranging from 0.1 to 0.269 g/L, chitin concentrations ranging from 5 to 13.45 g/L, and yeast extract concentrations ranging from 0.5 to 1.345 g/L. The proper range for each factor was determined based on the screening experimental results.

A central composite rotatable design of 47 experiments was composed, 32 of which corresponded to a complete factorial design (2^5^), ten experiments to star points (α ± 2.38), and five to the middle factor’s fields. The design of the experiments is given in Table 1. Chitinase activity was expressed as a function of independent variables (factors) by a polynomial equation of the second order:$$Y_{k}=\beta_{0}+\sum_{i=1}^{5}\beta_{i}X_{i}+\sum_{i=1}^{5}\beta_{ii}X_{i}^{2}+\sum_{i\neq j=1}^{5}\beta_{ij}X_{i}X_{j}$$
where *Y_k_* represents the measured response variables, *β*_0_ is a constant, and *β_i_*, *β_ii_*, and *β_ij_* are the linear, quadratic, and interactive coefficients of the model, respectively. *X_i_* and *X_j_* are the levels of the independent variables. Statistical analysis was performed using the software STATISTICA (version 7.0) for the experimental design and regression analysis of the experimental data. Student’s *t*-test was used to check the statistical significance of the regression coefficient and Fisher’s F-test was determined to adjust the second-order model equation at a probability (*p*) of 0.05. Model adequacy was evaluated using lack-of-fit, the coefficient of determination (R^2^), and the F-test value obtained from the analysis of variance (ANOVA). Model terms were selected based on the *p*-value (probability) with 95%. Regression analysis and three-dimensional response surface plots were plotted to determine the optimum conditions for chitinolytic activity (*Y*).

### 2.6. Partial Purification of Chitinase Activity of S614 Strain

Chitinase partial purification was performed at 4 °C. The S614 strain was cultivated under optimal conditions with the addition of colloidal chitin. Supernatant proteins were precipitated using 85% ammonium sulphate for a whole night. They were then recuperated by centrifuging at 14,000× *g*, dialyzed through a membrane with a molecular weight cutoff of 3.500 kDa in 10 mM Tris–HCl buffer (pH 6.5), and dissolved in the same buffer at 50 mM concentration. After a ten-fold concentration of the enzyme extract was added, aliquots were used for SDS-PAGE analysis or for the assessment of the antifungal activity.

### 2.7. SDS-PAGE Analysis of the Chitinase Extract

In accordance with Laemmli [26], a ten-fold concentrated chitinase extract was analyzed using 10% SDS-PAGE, with an LMW-SDS Marker Kit (GE Healthcare, Yvelines, France) serving as the standard. The extract containing chitinase was extracted, cleaned, stained, dehydrated in acetonitrile, dried in a vacuum centrifuge, and then tryptic digested at 37 °C for an entire night while 20 mg/mL of porcine trypsin (Sigma-Aldrich Inc., Saint-Louis, MO, USA) was present in a solution containing 40 mM ammonium bicarbonate, 10% acetonitrile, and 0.5% beta-octyl-D-glucoside. Using a MALDI Q-Tof Premier (Waters, Manchester, UK) tandem mass spectrometer, the resultant peptide mixture was examined. The MASCOT program (http://www.matrixscience.com, accessed on 15 March 2022) was then used to identify the chitinase protein using the NCBI non-redundant protein database. The data are indicative of separate gels in triplicate.

### 2.8. Antifungal Activity of Chitinase

Phytopathogenic fungi, including *Botrytis cinerea* M5 (broad bean) (National Institute of Agronomic Research of Tunisia, INRAT) and *Phoma medicaginis* Ph8 (alfalfa), provided by Dr. Naceur Djebali (CBBC, Borj-Cedria, Tunisia) were used as fungal indicators. Fungal growth was performed on potato dextrose agar (PDA, Fluka, Buchs, Switzerland) at 25 °C in the dark for seven days and then stored at 4 °C before use. The antifungal activities of different concentrations, 1.5 U, 1 U, and 0.5 U, of partially purified chitinase were tested by the disk diffusion method in which 0.5 U represents the minimum inhibitory concentration [25]. Disks of 4 mm diameter of 7-day-old test fungus culture were cut out and placed in the center of PDA plates. Extracts obtained in the presence (chitinase) or absence (control) of colloidal chitin under the optimized conditions were solubilized in 50 mM Tris–HCl buffer (pH 6.5) and then deposited on sterile 5 mm diameter Whatman paper discs at 2 cm from the fungal disc. The Petri dishes were incubated for a period appropriate for fungal growth (5 days for *Botrytis cinerea* and 10 days for *Phoma medicaginis*) at 25 °C. The diameters of the inhibition zone were measured to determine the antifungal activity of chitinase. Four replicates were considered for the antifungal activity of chitinase.

### 2.9. Chitinase Effect Using Detached Leaf Assay

Four-week-old *Medicago truncatula* plants were surface-sterilized by immersing detached leaves, free of wounds and diseases, in a 2% aqueous solution of sodium hypochlorite for three minutes. The leaves were then thoroughly cleaned with sterile distilled water, dried, and put on Petri plates with sterile filter paper soaked in water. Every leaflet had three needle pricks made on it. Leaf subjects were subjected to the following treatments: C represents untreated control leaves; P stands for leaves containing 10^6^ *P. medicaginis* conidia/mL; and Ch represents leaves concurrently injected with 10^6^ *P. medicaginis* conidia/mL and 0.5 U optimized crude chitinase. The leaves that received treatment were maintained at 25 °C and 95% relative humidity (RH) for ten days in the dark. Experiments were performed in triplicate, each consisting of 10 leaves excised from three plants. At the end of the incubation period, the percentage of the area of lesion was evaluated by determining the area by the Image J program according to the following formula: lesion area/total leaf area.

### 2.10. Statistical Analysis

The effect of chitinase treatment compared to the control, in vitro and *in planta*, was analyzed by one-way ANOVA/MANOVA using Statistica 5 software. Mean values were compared using the Duncan multiple range test at *p* 0.05.

## 3. Results

### 3.1. Identification and Phylogenetic Relationship between Burkholderia Isolate and GenBank Database

The S614 strain was isolated from soil in southern Tunisia and identified as the *Burkholderia contaminans* species based on 16S rRNA sequencing and BLASTn search results and deposited in GenBank under accession number MW922877. We constructed a phylogenetic tree to illustrate the relationships between isolate S614 based on 16S rRNA sequence identity to database sequences available in GenBank (Figure 1). Based on the 16S rRNA phylogenetic tree, pairwise comparison of the 16S rRNA sequences of *Burkholderia*-like strains revealed that their levels of identity ranged from 96.6 to 100%. In addition, the 16S rRNA of strain S614 showed 100% identity with the reference strain *B. contaminans* J2956, *B. paludis* strain MSh1, and 99.85% identity with the reference strain *B. lata* 383T. 

### 3.2. Chitinase Activity

The S614 strain showed high chitinolytic activity when cultivated on a chitin colloid agar plate. Extracellular chitinase activity was carried out in a degradation zone 25 mm in diameter. Production of chitinolytic activity by strain S614 reached 0.3 U/mL after 8 d of culture in a basal liquid culture medium containing colloidal chitin as the substrate (Figure 2, Table 1).

### 3.3. Model Fitting and Statistical Analysis

Experimental and predicted chitinase activities of *B. contaminans* strain 614 were both determined using the DNS method through central composite design (Table 1). Table 2 lists the regression coefficients and corresponding *p* values for different independent variables. The significance of coefficients and interaction strength were evaluated based on *p* values (variables with *p* < 0.05 were considered significant). Results indicated that all linear coefficients significantly influenced chitinase activity (*p* < 0.05). Moreover, only two independent variables including concentrations of ammonium sulphate (X1) and potassium phosphate (X2) had significant negative quadratic effects (*p* < 0.05) on chitinase activity response. A strong interaction effect was found between ammonium sulphate (X1) and potassium phosphate (X2); ammonium sulphate (X1) and colloidal chitin (X4); potassium phosphate (X2) and magnesium sulphate (X3); and potassium phosphate (X2) and colloidal chitin (X4). As shown in Table 2, the model was validated by lack-of-fit testing and the coefficient of multiple determination (R^2^). The variance analysis for the lack-of-fit test for chitinase activity response was non-significant (*p* > 0.05), suggesting that the model suited the experimental data well. Furthermore, the calculated coefficient of multiple determination (R^2^) of 0.852 indicated a strong correlation between response and separate variables.

### 3.4. Response Surface Analysis for Chitinolytic Activity of Burkholderia Contaminans Strain 614

After considering only the important factors (Table 2), the model obtained demonstrated the correlation between independent variables and their interaction with the predicted response of chitinase activity of *B. contaminans* strain 614, with a high correlation coefficient (R^2^ = 0.852) indicated by the following second-order polynomial equation:$$\begin{array}{cc} Y_{CA}=0.523853 & -0.08769\;X_{1}-0.06605\;X_{2}+0.057039\;X_{3}+0.099164\;X_{4}+0.090121\;X_{5}\\ & -0.026258\;X_{1}^{2}-0.024419\;X_{2}^{1}+0.0043137\;X_{1}X_{2}-0.049925\;X_{1}X_{4}\\ & -0.035612\;X_{2}X_{3}-0.026662\;X_{2}X_{4} \end{array}$$

As indicated in Table 2, the F test was applied to check the regression model’s validity. The Ficher (F-test) value of regression coefficients was greater than the tabular value (Fregression = 18.3934692 > Ftabulated (11.35; 0.05) = 2.08), and the *p* value was less than 0.0001, indicating that the model’s components had a significant influence on the chitinase activity response. The ratio of the mean square of lack-of-fit and pure error was lower than the tabulated value (Flack-of-fit = 3.50117873 < F tabular (31.4; 0.05) = 5.63), indicating that the lack-of-fit statistic was less significant (*p* > 0.05) than the raw error, owing to noise. As a result, the model is valid and adequate for estimating chitinolytic activity using any combination of variable values.

### 3.5. Optimization of Chitinase Activity of B. contaminans Strain 614

RSM was used to determine the levels of experimental factors to yield maximum chitinase activity of *B. contaminans* strain 614. As shown in Figure 3, three-dimensional response surfaces were plotted using the regression equation for the results of chitinase activity of *B. contaminans* strain 614 as a function of significant interaction between factors.

Figure 3a represents the effects of significant combined factors including (X1) ammonium sulphate and colloidal chitin (X4) when potassium phosphate (X2), magnesium sulphate (X3), and yeast extract (X5) were fixed at a −2, 1, and −1 levels, respectively. Chitinase activity exceeded 1.4 UA/mL with decreasing the ammonium sulphate concentration and increasing that of colloidal chitin. Figure 3b depicts the significant effects of combined factors including (X1) ammonium sulphate and potassium phosphate (X2) when colloidal chitin (X4), magnesium sulphate (X3), and yeast extract (X5) were fixed at a 1, 1, and −1 levels, respectively. The maximal chitinase production (higher than 1 UA/mL) was obtained at low concentrations of both salts (ammonium sulphate and potassium phosphate). In addition, the maximal value of chitinase production (higher to 1 UA/mL) was obtained with an increasing colloidal chitin concentration and a decreasing potassium phosphate concentration when ammonium sulphate (X1), magnesium sulphate (X3), and yeast extract (X5) were fixed at a −2, 1, and −1 levels, respectively (Figure 3c). Figure 3d reveals that this response could exceed 1.2 UA/mL with increasing the magnesium sulphate concentration (X3) and decreasing that of potassium phosphate (X2) when ammonium sulphate (X1), colloidal chitin (X4), and yeast extract (X5) were fixed at a −2, 1, and −1 levels, respectively.

Under optimal conditions, the measured chitinase activity was 1.15 AU/mL, which is consistent with the predicted value (YCA of 1.091 AU/mL). Interestingly, chitinase production increased 4-fold under optimized conditions compared to basal conditions (1.15 AU/mL versus 0.3 AU/mL in the basal control medium).

### 3.6. Partial Purification and Identification of Chitinase S614 Activity

In order to identify chitinase activity, strain S614 was cultured in the basal and optimized medium and then the secreted proteins were partially purified and concentrated 10 times. Three main bands were identified by SDS-PAGE (Appendix A) and each band was analyzed by tryptic digestion followed by sequencing using LC-ESI-MS/MS. The N1 band became more intense in the optimized protein extract than in the basal medium protein extract. Band N1 was identified as beta-N-acetyl hexosaminidase (90.1 kDa) based on its 14 peptide sequences: DRPGFALRRLTGDLYELTPQPGSVR, YVESLPADAQNNSTGNAPPVAARPDASR, RLPA DIATPGGYR, NFKHPATLR, SGGGYLTRDDYVSLVRYAAAHF, VEIIPEIDMPAHARAAVVTM, EARYQR, LLDPQDTSNL, TTVQFYDRR, EI AAMHADAQAPLHTWHYGGDEAK, IDLAAQDKPWAR, HANGPQDFSTR, GYYWGSHATDEYK, and IEGMQGQAWGEVMR (Figure 4). When submitted to BLAST (NCBI), these sequences revealed the highest degree of similarity with chitinases from different species of *Burkholderia*. Indeed, the 14 sequenced peptides demonstrated 100% similarity with the sequence of beta-N-acetyl hexosaminidase from *Burkholderia lata* 383 (accession number NR102890.1) (Table 3).

### 3.7. In Vitro and in Planta Antifungal Activity of Chitinase

The optimized and partially purified chitinase from S614 was tested for its antifungal activity by the disk diffusion method at a concentration of 0.5 U (Figure 5). Chitinase inhibited the mycelium growth of the phytopathogenic fungi *Botrytis cinerea* M5 and *Phoma medicaginis*, producing zones of inhibition with diameters of 16.7 mm and 12.3 mm, respectively. Based on the *in planta* test, infection by the pathogen alone caused significant damage (a percentage of lesions at the level of detached leaves of 53.05%), whereas the optimized chitinase appeared more effective in controlling *P. medicaginis* infectivity by reducing the detrimental effect of pathogen infection on *M. truncatula* leaves to only 14.18% (Figure 6).

## 4. Discussion

The present study aimed at assessing the effect of several factors in order to optimize a *Burkholderia contaminans* strain called S614, native to arid regions in Tunisia and potentially useful in biocontrol utilization, since it exhibits chitinase activity. The *B. cepacia* complex has been reported to encompass 24 closely related species called (Bcc), sharing high identity (>97.5%) in the 16S rRNA region [27]. An S614 isolate was identified as the species *Burkholderia contaminans* based on 16S rRNA sequencing and was deposited under the accession number MW922877 in GenBank. According to the phylogenetic tree, the *Burkholderia*-type strain S614 identity level ranged between 96.62 and 100% with *B. cepacia* complex reference strains. Furthermore, the 16S rRNA of strain S 614 showed 100% identity with the reference strain *B. contaminans* J2956 and 99.85% identity with the reference strain *B. lata* 383T. Several studies have shown that strain members of the Burkholderia group have been isolated as rhizospheric soil dwellers. *Burkholderia contaminans* is the dominant species of the *Burkholderia* community inhabiting low acidity (pH) soil, followed by *B. metallica*, *B. cepacia*, and *B. stagnalis*, respectively [28]. All species belonging to the *Burkholderia cepacia* (Bcc) complex are considered opportunistic pathogens that also display biological features beneficial for agricultural applications and biodegradation purposes [29]. In addition, some environmental isolates of the Bcc group have been reported to synthesize antimicrobial compounds that are not pathogenic [30].

Recently, various chitinase-producing bacteria have been found to completely hydrolyze chitin powder, yielding GlcNAc at high levels [28]. The enzymatic degradation process for generating GlcNAc may be more attractive than acid hydrolysis due to its moderate conditions, simple substrate preparation, and high production yield [7]. The strain S614 showed extracellular chitinolytic activity, revealed by a degradation zone of 25 mm diameter and the production of enzymatic activity (0.3 U/mL) in a basal liquid culture medium containing 0.5% colloidal chitin. It has been reported that *Burkholderia metallica*, *Burkholderia stagnalis*, and *B. contaminans* strains were able to produce hydrolytic enzymes such as the following: chitinase, amylase, cellulase, and protease [28,31].

In order to improve the enzyme production of *B. contaminans* strain 614, its experimental and predicted chitinase activities obtained from a central composite plane were estimated by the DNS method. RSM was also used to determine the levels of experimental factors needed to achieve maximum chitinase activity. In our RSM study, the model proved to be valid and suitable for predicting chitinolytic activity under any combination of variable values. The *p*-value was used to measure the significance of each element, which is essential for understanding the interactions between variables. A lower *p*-value suggests a more significant coefficient [32]. The central composite experimental design is considered as a strong method to identify the best levels of important factors and their interactions in chitinase production by microorganisms. [32]. Based on the MSR, the optimal concentrations of the independent variables marked with a dot on the surface were as follows: ammonium sulfate (X1) (3.5 g/L), potassium phosphate (X2) (0.5 g/L), magnesium sulfate (X3) (0.2 g/L), colloidal chitin (X4) (10 g/L), and yeast extract (X5) (0.5 g/L). In this optimal response surface (ORS) method, the levels of the experimental factors were used to determine the maximum chitinase activity of *B. contaminans* strain 614. Plotting the response surface curve allowed for (i) a better understanding of the interactions of the variables and (ii) the identification of the optimal level of each variable from the perspective of maximizing chitinase production. Hence, the optimal production of chitinase was obtained at a high concentration of colloidal chitin, which is in agreement with Philip et al. [21], suggesting that colloidal chitin is the most appropriate substrate for the highest yields of chitinase activity. Garima et al. [23] showed that a higher concentration of colloidal chitin (15 g/L) enhanced the optimal secretion of chitinase by *B. cereus* GS02. Chitin degradation is a regulated process, and Chitinases are adaptive (inducible) enzymes and are regulated by a repressor/inducer system. The colloidal chitin substrate is an inducer, while easily assimilating carbon sources such as glucose act as a repressor [22]. Without the inducer chitin, no chitinase extracellular production was observed [33]. However, variability in chitinase activity between various strains and species correlates with variation in colloidal chitin concentrations [23]. In contrast, additional research has indicated that *Bacillus* sp. A 14 has the capability to produce chitinases even without the need for an inducer like chitin in its natural state [34]. In *Vibrio furnissii*, the basal produced extracellular chitinase solubilizes colloidal chitin to soluble oligosaccharide products. The latter penetrate the outer membrane and are degraded in the periplasmic space to GlcNAc oligomers [35]. Chitin hydrolytic enzymes can be induced by products of the chitin hydrolyse, such as (GlcNAc)_2_ oligomer principally [36,37], or by GlcNAc [38], depending on the microorganism. Moreover, in *Vibrio furnissii*, a surface chitin-binding protein interacts with a cytoplasmic membrane-anchored regulator, and the (GlcNAc)_2_ binding induces a conformational change in the repressor, inducing chitinase gene via the phosphorylation of a cytoplasmic regulator [36].

It is crucial to improve enzyme production by optimizing trace elements such as magnesium sulfate, potassium dihydrogen phosphate, and dipotassium hydrogen phosphate incorporated into the culture medium, as these elements play an important role in microbial growth [39]. Variations in chitinase synthesis in the presence of different salts comprising calcium, iron, mercury, magnesium, and zinc are also reported in other research [40,41].

Based on RSM data, the maximum chitinase production could be obtained at low levels of ammonium sulphate and potassium phosphate and high levels of (magnesium sulphate and colloidal chitin), as reflected by the fact that chitinase production increased with a decreased PO_4_ (K_2_HPO_4_) concentration. Meriem et al. [32] showed that PO_4_ addition at a low level increased the chitinase production of *Streptomyces griseorubens* C9, whereas high levels of PO_4_ led to a slight decrease in chitinase production, which may be related to the negative effect of enhancing cytoplasmic osmotic pressure. Potassium ions have been reported to play an essential role in the physiological activities of microbial cells besides their involvement in the synthesis and regulation of primary and secondary metabolites. However, at high concentrations, inorganic phosphate may alter protein accumulation through the resulting enzyme production [39].

The positive effect of MgSO_4_ on chitinase production by different bacterial strains such as *Bacillus* sp. and *Paenibacillus* sp. is known to be notably explained by the role of Mg^2+^ in cell growth, enzyme production, and stability [42]. Similarly, Qu et al. [7] showed that metal ions such as Cu^2+^, Hg^2+^, and Co^2+^ inhibit most chitosanases and chitinases, while metal ions such as Mg^2+^, Ca^2+^, and Mn^2+^ activate them. Interestingly, the latter promotes chitosan hydrolysis by AoNagase, but inhibits chitin hydrolysis. Nitrogen sources also affect chitinase production as data inferred from the present study show that low levels of ammonium sulfate had a positive effect on chitinase production. This was confirmed by Kotb et al. [43], who reported that ammonium nitrate, ammonium acetate, and sodium nitrate had no effect on the enhancement of chitinase productivity by *Streptomyces* sp. strain ANU627713.

After maximum chitinase activity was determined, chitinase identification was performed using MASCOT software version 2.6 (http://www.matrixscience.com, accessed on 15 March 2022). The peptide sequences of this protein covered approximately 20% of the beta-N-acetylhexosaminidase protein sequence responsible for colloidal chitin degradation in GlucNac. The final hydrolytic reaction of beta-N-acetylhexosaminidase yielded pure GlcNAc without any by-products, indicating wide applicability for the enzymatic production of this highly valued chemical [44].

In previous studies, several substrates have been found to be degraded due to the β-N-acetylhexosaminidase in the GH 20 family. β-N-acetylhexosaminidase of the GH 20 family of *Trichoderma reesei* uses only p-nitrophenyl-N-acetyl-beta-D-glucosaminide as a substrate [45], whereas the enzyme produced by *Lentinula edodes* can degrade pNP-GlcNAc, p-nitrophenyl-N-acetyl-beta-D-galactosaminide, chitin, colloidal chitin (46.3 U/mg), and mechanically crushed chitin (39.9 U/mg) to GlcNAc [46]. In the same way, Qu et al. [7] demonstrated that β-N-acetylhexosaminidase is able to hydrolyze not only pNP-GlcNAc, but colloidal chitin and chitosan.

According to the *in planta* test, following infection with the pathogen *P. medicaginis*, the optimized chitinase (β-N-acetylhexosaminidase) was found to be very effective in reducing the harmful effect of infection on *M. truncatula* leaves. To the best of our knowledge, this is the first study in the context of biocontrol against *P. medicaginis* carried out with *Burkholderia* and chitinase on the legume model *M. truncatula*. Comparable results were reported by Tagele et al. [31], who noted that *Burkholderia* strain KNU17BI1 was more effective in (i) inhibiting mycelial growth and sclerotic germination of *Rhizoctonia solani*, and (ii) the biological control of various economically important plant fungal pathogens. This strongly suggests that this capacity can be ascribed to the ability of *Burkholderia* KNU17BI1 to produce antifungal metabolites, nutrient competition, and proteolytic enzymes. Ren et al. [47] also mentioned that the non-pathogenic strain *Burkholderia. pyrrocinia* JK-SH007 showed biocontrol abilities against *Phomopsis macrospora*, *Cytospora chrysosperma*, and *Fusicoccum aesculi* in relationship to the production of extracellular hydrolytic enzymes (β-1, 3-glucanases, chitinases, and proteases).

## 5. Conclusions

As a whole, following the purification and characterization of chitinase from *B. contaminans* S614, we show that RSM is effective in optimizing culture medium components and increasing chitinase synthesis by *B. contaminans* S614. Given its significant effectiveness in both in vitro and in vivo biocontrol of phytopathogenic fungi, *B. contaminans* S614 could thus be used to improve the efficiency of industrial chito-oligomeric processes and develop an effective microbiota for the biocontrol of phytopathogens in sustainable agriculture.

## Figures and Tables

**Figure 1 microorganisms-12-01580-f001:**
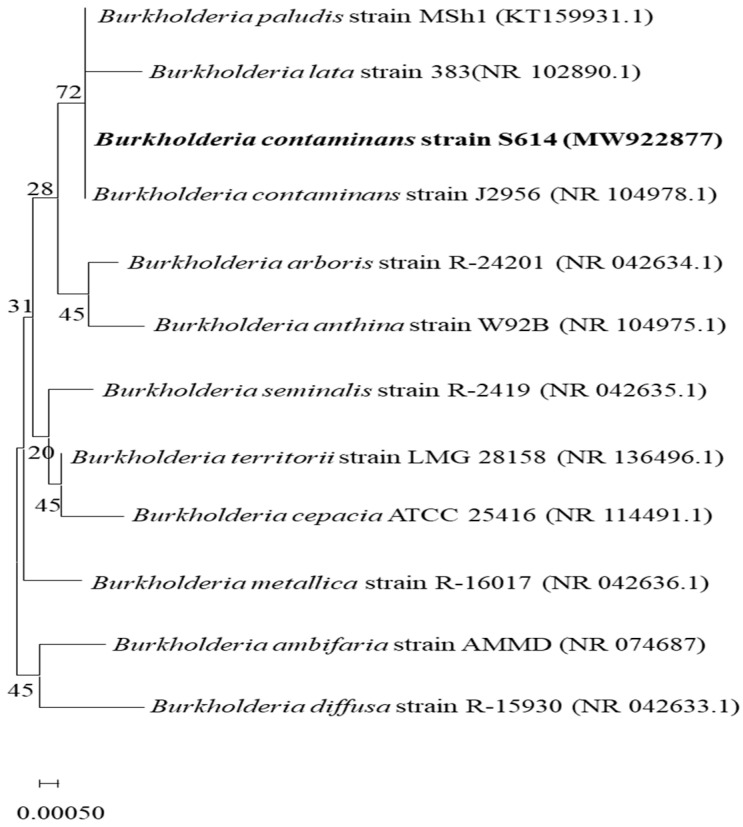
Neighbor-joining tree constructed on the basis of 16S rRNA of 1300 bp fragment showing the relation between strain S614 and other different related species. The GenBank accession number of each species is given after strain type name.

**Figure 2 microorganisms-12-01580-f002:**
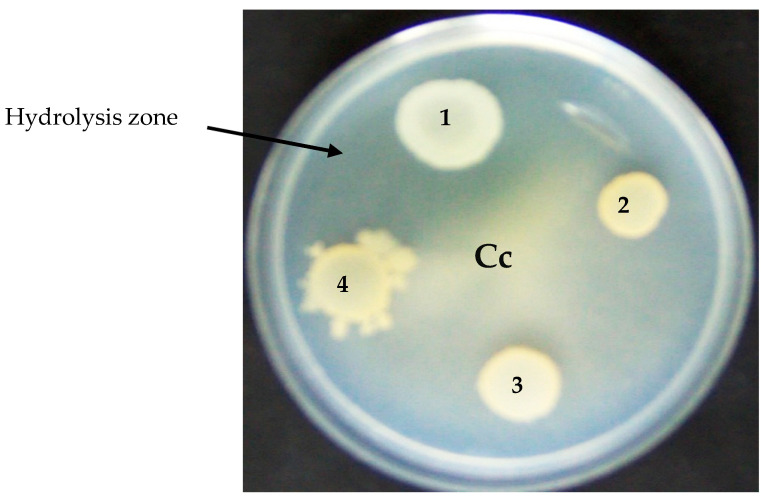
Detection of chitinolytic activity from S614 strain on colloidal chitin plate. (1) S614 strain, (2) *Bacillus subtilis* L193, (3) *Bacillus subtilis* L32, (4) *Bacillus velezensis* L194 non-chitinolytic strains, and (Cc) Agar medium containing colloidal chitin. The arrow shows the hydrolysis zone of colloidal chitin.

**Figure 3 microorganisms-12-01580-f003:**
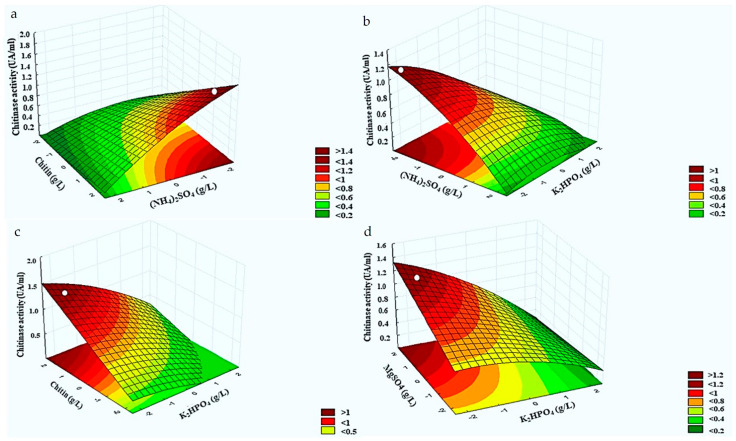
Response surface curves showing the effect of the interaction between relevant factors, (NH_4_)_2_SO_4_ (X1), K_2_HPO_4_ (X2), MgSO_4_ (X3), and colloidal chitin (X4), on chitinase production by *B. contaminans* S614. (**a**) Interaction X1X4, (**b**) interaction X1X2, (**c**) interaction X4X1, and (**d**) interaction X3X1.

**Figure 4 microorganisms-12-01580-f004:**
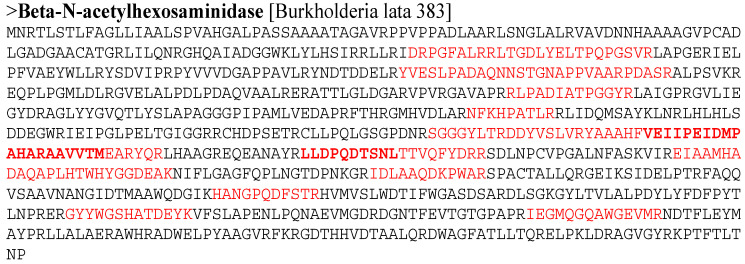
Location of 14 peptides (red text) of beta-N-acetylhexosaminidase from strain S614 in beta-N-acetylhexosaminidase from *Burkholderia lata* 383 under accession number NR102890.1.

**Figure 5 microorganisms-12-01580-f005:**
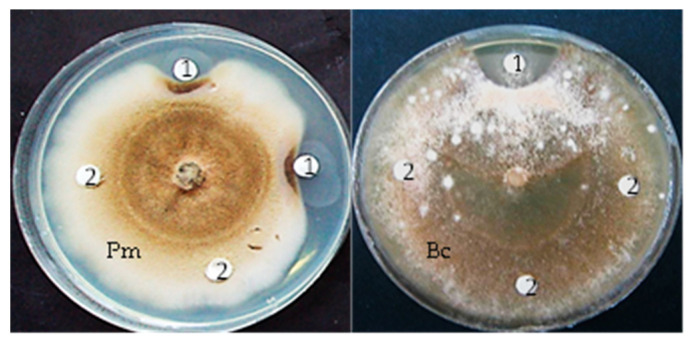
Antifungal activity of chitinase extract from S614 strain against *Phoma medicaginis* (Pm) and against *Botrytis cinerea* (Bc): (1) 0.5 U chitinase activity unit and (2) negative control (buffer).

**Figure 6 microorganisms-12-01580-f006:**
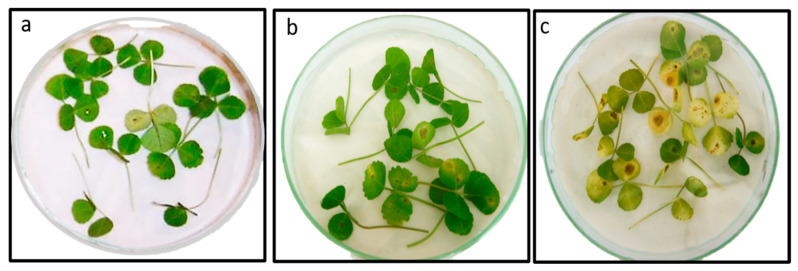
Protective effect of chitinase of *B contaminans* S614 against *Phoma medicaginis* infection in *Medicago truncatula*. (**a**) Untreated control leaves, (**b**) leaves pretreated with S614 chitinase solution (1.15 U and inoculated with *P. medicaginis* conidia (10^6^/mL) (lesion %: 14.18 ± 10), and (**c**) leaves inoculated with *P. medicaginis* conidia (10^6^/mL) (lesion %: 53.05 ± 18).

**Table 1 microorganisms-12-01580-t001:** Coded levels and condition runs with the experimental and predicted values used in central composite design (CCD) for chitinase activity response.

Exp N°	Independent Variables				Experimental Value (Y_CA_)		Predicted Values (Y_CA_)
	(NH_4_)_2_SO_4_ (g/L) (X1)	K_2_HPO_4_ (g/L) (X2)	MgSO_4_ (g/L) (X3)	Chitin (g/L) (X4)	Yeast Extact (g/L) (X5)	Chitinase Activity (UA/mL)	Chitinase Activity (UA/mL)
**1**	7 (−1)	1 (−1)	0.1 (−1)	5 (−1)	0.5 (−1)	0.3848	0.311541
**2**	14 (1)	1 (−1)	0.1 (−1)	5 (−1)	0.5 (−1)	0.1720	0.149732
**3**	7 (−1)	2 (1)	0.1 (−1)	5 (−1)	0.5 (−1)	0.1510	0.217698
**4**	14 (1)	2 (1)	0.1 (−1)	5 (−1)	0.5 (−1)	0.2290	0.228439
**5**	7 (−1)	1 (−1)	0.2 (1)	5 (−1)	0.5 (−1)	0.4284	0.496843
**6**	14 (1)	1 (−1)	0.2 (1)	5 (−1)	0.5 (−1)	0.3114	0.335034
**7**	7 (−1)	2 (1)	0.2 (1)	5 (−1)	0.5 (−1)	0.3428	0.260550
**8**	14 (1)	2 (1)	0.2 (1)	5 (−1)	0.5 (−1)	0.3148	0.271291
**9**	7 (−1)	1 (−1)	0.1 (−1)	10 (1)	0.5 (−1)	0.6412	0.663044
**10**	14 (1)	1 (−1)	0.1 (−1)	10 (1)	0.5 (−1)	0.3308	0.301536
**11**	7 (−1)	2 (1)	0.1 (−1)	10 (1)	0.5 (−1)	0.3646	0.462551
**12**	14 (1)	2 (1)	0.1 (−1)	10 (1)	0.5 (−1)	0.2268	0.273592
**13**	7 (−1)	1 (−1)	0.2 (1)	10 (1)	0.5 (−1)	0.8804	0.848346
**14**	14 (1)	1 (−1)	0.2 (1)	10 (1)	0.5 (−1)	0.3128	0.486838
**15**	7 (−1)	2 (1)	0.2 (1)	10 (1)	0.5 (−1)	0.4086	0.505403
**16**	14 (1)	2 (1)	0.2 (1)	10 (1)	0.5 (−1)	0.3086	0.316444
**17**	7 (−1)	1 (−1)	0.1 (−1)	5 (−1)	1 (1)	0.4472	0.491783
**18**	14 (1)	1 (−1)	0.1 (−1)	5 (−1)	1 (1)	0.2432	0.329974
**19**	7 (−1)	2 (1)	0.1 (−1)	5 (−1)	1 (1)	0.4446	0.397939
**20**	14 (1)	2 (1)	0.1 (−1)	5 (−1)	1 (1)	0.3478	0.408681
**21**	7 (−1)	1 (−1)	0.2 (1)	5 (−1)	1 (1)	0.6294	0.677085
**22**	14 (1)	1 (−1)	0.2 (1)	5 (−1)	1 (1)	0.7018	0.515276
**23**	7 (−1)	2 (1)	0.2 (1)	5 (−1)	1 (1)	0.4266	0.440791
**24**	14 (1)	2 (1)	0.2 (1)	5 (−1)	1 (1)	0.499	0.451533
**25**	7 (−1)	1 (−1)	0.1 (−1)	10 (1)	1 (1)	0.6424	0.843286
**26**	14 (1)	1 (−1)	0.1 (−1)	10 (1)	1 (1)	0.5796	0.481777
**27**	7 (−1)	2 (1)	0.1 (−1)	10 (1)	1 (1)	0.7054	0.642793
**28**	14 (1)	2 (1)	0.1 (−1)	10 (1)	1 (1)	0.5526	0.453834
**29**	7 (−1)	1 (−1)	0.2 (1)	10 (1)	1 (1)	1.1714	1.028588
**30**	14 (1)	1 (−1)	0.2 (1)	10 (1)	1 (1)	0.6488	0.667079
**31**	7 (−1)	2 (1)	0.2 (1)	10 (1)	1 (1)	0.7022	0.685645
**32**	14 (1)	2 (1)	0.2 (1)	10 (1)	1 (1)	0.5228	0.496686
**33**	2.17 (−2.38)	1.5 (0)	0.15 (0)	7.5 (0)	0.75 (0)	0.6608	0.583884
**34**	18.83 (2.38)	1.5 (0)	0.15 (0)	7.5 (0)	0.75 (0)	0.1020	0.166749
**35**	10 (0)	0.31 (−2.38)	0.15 (0)	7.5 (0)	0.75 (0)	0.5774	0.542832
**36**	10 (0)	2.69 (2.38)	0.15 (0)	7.5 (0)	0.75 (0)	0.2062	0.228600
**37**	10 (0)	1.5 (0)	0.031 (−2.38)	7.5 (0)	0.75 (0)	0.4750	0.388192
**38**	10 (0)	1.5 (0)	0.269 (2.38)	7.5 (0)	0.75 (0)	0.6112	0.659514
**39**	10 (0)	1.5 (0)	0.15 (0)	1.55 (−2.38)	0.75 (0)	0.2690	0.288000
**40**	10 (0)	1.5 (0)	0.15 (0)	13.45 (2.38)	0.75 (0)	0.8450	0.759707
**41**	10 (0)	1.5 (0)	0.15 (0)	7.5 (0)	0.155 (−2.38)	0.3652	0.309509
**42**	10 (0)	1.5 (0)	0.15 (0)	7.5 (0)	1.345 (2.38)	0.5530	0.738198
**43**	10 (0)	1.5 (0)	0.15 (0)	7.5 (0)	0.75 (0)	0.5454	0.523853
**44**	10 (0)	1.5 (0)	0.15 (0)	7.5 (0)	0.75 (0)	0.4872	0.523853
**45**	10 (0)	1.5 (0)	0.15 (0)	7.5 (0)	0.75 (0)	0.4910	0.523853
**46**	10 (0)	1.5 (0)	0.15 (0)	7.5 (0)	0.75 (0)	0.5528	0.523853
**47**	10 (0)	1.5 (0)	0.15 (0)	7.5 (0)	0.75 (0)	0.6120	0.523853

**Table 2 microorganisms-12-01580-t002:** Regression coefficients and analysis of variance (ANOVA) of the predicted second-order polynomial model for chitinase activity response.

Y_CA_ (R^2^ = 0.8525)					
Terms	Regression Coefficients ^a^	Standard Error	*t* Value	*p* Value	
β_0_	0.523853	0.012205	42.9200	>0.0001	
β_1_	−0.087692	0.007799	−11.2436	0.000356	
β_11_	−0.026258	0.006923	−3.7926	0.019226	
β_2_	−0.066059	0.007799	−8.4699	0.001065	
β_22_	−0.024419	0.006923	−3.5271	0.024295	
β_3_	0.057039	0.007799	7.3133	0.001860	
β_4_	0.099164	0.007799	12.7145	0.000220	
β_5_	0.090121	0.007799	11.5550	0.000320	
β_12_	0.043138	0.009074	4.7540	0.008944	
β_14_	−0.049925	0.009074	−5.5021	0.005321	
β_23_	−0.035613	0.009074	−3.9247	0.017179	
β_24_	−0.026663	0.009074	−2.9384	0.042458	
Source of variation	SS	DF	MS	*F* Value	*p* Value
Regression	1.714030	11	0.15582087	18.3934692	<0.0001
Residuals	0.296504	35	0.00847153		
Lack of fit	0.2859647	31	0.00922467	3.50117873	0.1149
Pure error	0.01053893	4	0.00263473		
Total	2.010533	46			

^a^: Terms with *p* < 0.05 correspond to significant independent variables. SS: sum of squares; DF: degree of freedom; MS: mean square; *F* value: Fisher value.

**Table 3 microorganisms-12-01580-t003:** Characterization of β-N-acetylhexosaminidase S614 after tryptic digestion and LC-ESI-MS/MS analysis.

Name	Link	Strain	MW	Peptide Carte	% Recovery MS	Peptide Sequence
beta-N-acetylhexosaminidase	Q39CS2_BURS3	*Burkholderia lata* (strain 383)	90,135	14	20	DRPGFALRRLTGDLYELTPQPGSVR YVESLPADAQNNSTGNAPPVAARPDASR RLPADIATPGGYR NFKHPATLR SGGGYLTRDDYVSLVRYAAAHF VEIIPEIDMPAHARAAVVTM EARYQR LLDPQDTSNL TTVQFYDRR EIAAMHADAQAPLHTWHYGGDEAK IDLAAQDKPWAR HANGPQDFSTR GYYWGSHATDEYK IEGMQGQAWGEVMR

## Data Availability

The Datasets presented in this article are not readily available due to time limitations. Requests to access the datasets should be directed to Imen Ben Slimene Debez.

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
