# Peer review of "Response Surface Methodology-Based Optimization of the Chitinolytic Activity of Burkholderia contaminans Strain 614 Exerting Biological Control against Phytopathogenic Fungi"

_microorganisms, 2024, doi:10.3390/microorganisms12081580_

Round 1

Reviewer 1 Report

Comments and Suggestions for Authors

The manuscript reported chitinase activity of Burkholderia contaminans S614 strain could be enhanced by a combination of factors according to the result of response surface methodology (RSM) with a central composite plane (CCD) analysis. This enzyme was identified by SDS-PAGE analysis, which was further tested for its antifungal activities against two important phytopathogenic fungi. The overall research design was reasonable, results were clearly presented. Discussion was sound. However, there are a number of drawbacks which need to be improved, and language errors were detected.

 1. In terms of In vitro and in planta antifungal activity of chitinase, I do not think images of inoculation are sufficient to support the conclusion. Experimental/biological replicates and statistical analysis of lesion incidence/area are required to support the conclusion.

 2. Strain name should be italic, e.g. Line 277, line 389 B. contaminans should be Italic, please check throughout the ms for language problems.

 3. Name of chemical compounds needs to be edited, for instance, Line 138 K2HPO4, MgSO4, 2 and 4 should be lower case, check all chemical names.

 4. Method for phylogenetic analysis needs to be added in Method Section

 5. Regarding in planta antifungal activity of chitinase, I wonder why antifungal activity of chitinase against Botrytis cinerea was not performed? Can this experiment be supplemented?

 The authors need to address the above comments before the ms can be accepted for publication.

Comments on the Quality of English Language

The overall writing is fine, except a small number of language problems such as Italic for Species name, lower case for chemical compounds. 

Reviewer 2 Report

Comments and Suggestions for Authors

The manuscript describes the optimization of the chitinase production by Burkhoderia contaminans strain 614 using experimental design and RSM. According to the authors, the enzymatic production was improved and the colloidal chitin was an important factor for this. Additionally, the enzyme inhibited the fungal development of two pathogenic strains, collaborating with the reduction of the infection of the plant leaves. In fact, experimental design is a powerful tool to improve the production of biomolecules by microorganisms, including enzymes. In this context, there are many works describing the improvement of enzyme production by bacteria including the chitinase. Under this view, any novelty is presented.

1 1. The English language should be revised by a Professional Proofreading Service. There are some sentences that need restructuring. Additionally, minor errors need attention. The biological nomenclature must be revised. Some genera are not as italic, specially in the References section. The g force should be italicized. The “spp.” Is not italic (line 46). Replace “Aminooligosaccharides” by “aminooligosaccharides” (line 49). The same idea presented in the lines 55-59 is repeated in the lines 66-70. The chemical formulas should be corrected through the text (for example, line 108; caption of the figure 3). Delete the “n” (line 208).

22. Improve the information on methodology for different parts.

22.1  There is no information on the Culture collection where the strain 614 is deposited. Please, provide including the access ID. It is not also clear if the strain was collected specifically for this work or previously. If it was collected for this investigation, provide all information on isolation procedure. If not, add reference on the isolation.

22.2 Add information on the sequence of the primers used or the reference where they were obtained.

22.3  For the experimental design, the authors considered 5 factors for the CCD. In this case, the number of experiments is great and, generally, far away of the optimized conditions. The best way is to perform a fractioned design 25-1to observe the influence of each variable. After this, to perform a CCD with 2 or 3 variables. Why the authors chosen the CCD with 5 variables?

33. It was mentioned in the caption of the figure 2 “The arrow showed the hydrolysis zone of colloidal chitin”. However, there is no arrow in the figure. Were the non-chitinolytic strains identified? If yes, mention each one in the caption.

44. Line 241. It was mentioned that the enzyme activity was 0.3 U/mL using basal medium with reference to the Table 1. However, it is not clear what can be considered as basal medium. Is the first run presented in the table? Aren't they all basal media but with different concentrations of nutrients?

55. The second-order polynomial equation is not legible. Line 337 – the Eq. (5) mentioned was lost. The R2 value was 0.852. This value is acceptable for the screening of variables, but for the CCD it should be at least 0.9.

66. Three protein bads were obtained through SDS-PAGE. These bands were digested with trypsin and the tryptic fragments were analyzed using mass spectrometry. Why only the result on the band 1 was shown? What are the other 2 protein bands?

77. The antifungal test considered only 0.5 U of the chitinase. It is interesting to test other enzyme concentrations to determine the minimal inhibitory concentration. Why only one disc with chitinase was used for Bc inhibition? Why the authors used agar diffusion if there are more effective methods, as the determination of the cell viability, to analyze the inhibitory activity of the enzyme on fungi?

88. The extract containing the enzyme was precipitated with ammonium sulfate and centrifuged using a membrane with 3.500 kDa. These procedures were considered as purification. Is it really purification or only concentration? How many times the enzyme was purified? Show the electrophoresis with crude extract and with “purified” fraction for comparison.

99. It was mentioned that the colloidal chitin is the inducer for the production of chitinase by Burkholderia strain (line 522). But this aspect needs reflection. Chitin is a great molecule to be transported to inside the cell. Is it possible that the bacterial cell produces minimal levels of chitinase constitutively? Is yes, the initial hydrolysis of the colloidal chitin by the enzyme could produce the true inducer. Discussion on this aspect is necessary.

110. Line 545. Humicola is not bacteria. It is a fungus. Correct.

Comments on the Quality of English Language

See comments and sugestions for authors.

Reviewer 3 Report

Comments and Suggestions for Authors

Review Report: microorganisms-3032981

The pursuit of environmentally friendly and sustainable methods for controlling phytopathogenic fungi has led to increased interest in biological control agents. Among the promising candidates is the species Burkholderia cepacia, known for its production of hydrolytic enzymes, such as chitinases. These enzymes degrade chitin, a major component of fungal cell walls, thereby inhibiting fungal growth. This study focuses on a specific strain, Burkholderia contaminans S614, isolated from southern Tunisian soil, and evaluates its potential as a biocontrol agent through chitinase production.

The Burkholderia contaminans strain S614 was isolated from soil samples collected in southern Tunisia. Its ability to produce chitinase was confirmed through biochemical assays. The enzyme responsible for chitinase activity was further identified as beta-N-acetylhexosaminidase, with a molecular weight of 90.1 kDa, based on peptide sequence homology to Burkholderia lata strain 383. Response Surface Methodology (RSM) with a Central Composite Design (CCD) was employed to optimize the production of chitinase. Five factors were considered: colloidal chitin, magnesium sulfate, dipotassium phosphate, yeast extract, and ammonium sulfate. This statistical approach enabled the determination of the optimal conditions for maximal chitinase activity.

Growth in the optimized medium resulted in a threefold increase in chitinase activity for the B. contaminans S614 strain. This significant enhancement underscores the efficacy of the RSM optimization process. The chitinase produced by B. contaminans S614 demonstrated substantial antifungal activity against two phytopathogenic fungi: Botrytis cinerea M5 and Phoma medicaginis Ph8. In vitro assays confirmed the inhibitory effects of the enzyme on fungal growth. The crude enzyme extract from strain S614 was further tested for its biocontrol potential on detached leaves of Medicago truncatula. The results indicated a marked reduction in damage caused by Phoma medicaginis, highlighting the practical applicability of this biocontrol strategy in agricultural settings.

The findings from this study provide compelling evidence for the biotechnological and agricultural potential of Burkholderia contaminans S614 as a biocontrol agent. The strain's ability to produce high levels of chitinase under optimized conditions, coupled with its effective inhibition of key phytopathogenic fungi, positions it as a viable alternative to chemical fungicides.

Comments to authors

Shortcomings in Materials and Methods

a)       Lack of Detailed Description for Strain Isolation and Maintenance:

·         The methods for isolating strain S614 from soil samples and the subsequent identification of the specific colonies exhibiting chitinase activity are not thoroughly described. Detailed isolation protocols, including the type of soil, sample preparation, and screening process, should be provided.

·         The storage conditions mention two methods (−80°C in glycerol and LB agar at 4°C), but there is no information on how these methods were validated for maintaining strain viability and activity.

b)      Insufficient Information on Culture Conditions:

·         The minimal medium composition is provided, but details on the preparation process, sterility measures, and exact incubation conditions (e.g., specific temperature and pH control measures during incubation) are lacking.

·         There is no mention of whether preliminary experiments were conducted to determine the optimal incubation period of 8 days or the chosen rotation speed of 150 RPM.

c)       Inadequate Explanation of Experimental Design:

·         The central composite design (CCD) methodology is introduced, but there is no clear justification for the selected range of factor levels or the choice of these specific factors. The rationale for choosing concentrations and their respective ranges should be elaborated.

·         The statistical methods and software used (STATISTICA version 7.0) are mentioned, but there is a lack of clarity on how the initial screening experiments informed the selection of ranges for each factor.

d)      Chitinase Activity Assay Lacks Specificity:

·         The preparation of the colloidal chitin solution refers to another study (reference [24]), but specific steps are not detailed within this manuscript. A concise summary of the preparation procedure should be included.

e)      Partial Purification Protocol is Ambiguous:

·         The partial purification process mentions the use of ammonium sulfate precipitation and dialysis but lacks specifics on the concentration steps, dialysis buffer composition, and volume reductions.

f)        Antifungal Activity Assessment Lacks Detailed Methodology:

·         The disk-diffusion method is referenced (reference [25]), but essential details such as the preparation of fungal cultures, the incubation conditions for the antifungal assay, and the criteria for measuring inhibition zones are not provided.

·         There is no information on how the results were quantified and analyzed statistically, nor are the controls (e.g., disks with no enzyme or inactive enzyme) described.

By addressing these shortcomings in M&M section of the MS, the manuscript will provide a more comprehensive and detailed account of the materials and methods used, enhancing the reproducibility and reliability of the findings. Based on a review I recommend a major revision.

Round 2

Reviewer 2 Report

Comments and Suggestions for Authors

The manuscript was improved considering the reviewer comments. In spite of this, my main criticism remains the lack of novelty. This point should be clearly presented. Other points still to be clarified/corrected in the document.

1. Considering the table 2, the authors highlighted only the enzyme activity with the basal medium (0.3 UA/mL). It will be interesting the mention of the run 29, where the best enzyme activity was obtained (1.17 UA/mL). The basal medium mentioned by authors presents (NH4)SO4, KH2PO4, NaCl, MgSO4 and yeast extract (line 117), and colloidal chitin added. According to this, at line 279, it is important to say modified basal medium, because NaCl is not present.

2. The authors explained in the response to reviewer why they applied a CCD with five factors. It is ok, but the information should be added in the manuscript text. The authors need to say that preliminary studies were done using eight factors and five were selected for the present study (data not shown).

3. The figure showing the SDS-PAGE profile (as shown in the response to reviewer) should be included in the manuscript as supplementary. Why the purified fraction was not loaded in the SDS-PAGE?

4. In the Material and Methods section add information on the use of different concentrations of the purified enzyme in the antifungal test and that 0.5 U was found as minimal inhibitory concentration, as mentioned in the response to reviewer.

Minor:

Line 41 – Add comma after the “gloesporioides”.

Line 43 – The “spp”. is not italicized.

Line 121 – The “g” is italicized.

Line 139 – Add the end point after “[24]”.

Line 190 – Correct “14,000 x g”.

Line 197 – Replace “induced” by “containing”.

Line 246 – The “Burkholderia” should be italicized.

Table 1 – Replace “l” by “L” and “ml” by “mL”.

Lines 327 and 330 – The “R2” should be “R2”.

Item 3.4 – The equation is not legible.

Figure 4 – Add information on the red text and the bold text in the legend of this figure.

Figure 6 – The “Medicago trunculata” should be italicized.

Lines 503 and 504 – Correct “106”.

Line 531 – Replace “ml” by “mL”.

Line 623 – The “Cytospora” should be italicized.

Standardize “days” or “d” through the text.

Comments on the Quality of English Language

See comments for authors.

Author Response

Please see our responses as attachment
